# How do Large Language Models Handle Multilingualism?

**Yiran Zhao**[1,2†]  **Wenxuan Zhang**[2,3‡]  **Guizhen Chen**[2,4§]  **Kenji Kawaguchi**[1]  **Lidong Bing**[2,3]

[1] National University of Singapore    [2] DAMO Academy, Alibaba Group, Singapore
[3] Hupan Lab, 310023, Hangzhou, China    [4] Nanyang Technological University, Singapore

## Abstract

Large language models (LLMs) have demonstrated impressive capabilities across diverse languages. This study explores how LLMs handle multilingualism. Based on observed language ratio shifts among layers and the relationships between network structures and certain capabilities, we hypothesize the LLM's multilingual workflow (`MWork`): LLMs initially understand the query, converting multilingual inputs into English for task-solving. In the intermediate layers, they employ English for reasoning and incorporate multilingual knowledge with self-attention and feed-forward structures, respectively. In the final layers, LLMs generate responses aligned with the original language of the query. To verify `MWork`, we introduce Parallel Language-specific Neuron Detection (`PLND`) to identify activated neurons for inputs in different languages without any labeled data. Using `PLND`, we validate `MWork` through extensive experiments involving the deactivation of language-specific neurons across various layers and structures. Moreover, `MWork` allows fine-tuning of language-specific neurons with a small dataset, enhancing multilingual abilities in a specific language without compromising others. This approach results in an average improvement of 3.6% for high-resource languages and 2.3% for low-resource languages across all tasks with just 400 documents.[1]

## 1 Introduction

Recent advancements in large language models (LLMs) (OpenAI, 2023; Touvron et al., 2023; Team et al., 2023) have dramatically transformed the field of natural language processing (NLP). Thanks to the extensive pretraining on massive corpora mixed with different languages, these models demonstrate remarkable capabilities in understanding and generating text across multiple languages (Huang et al., 2023; Zhang et al., 2023a; Zhao et al., 2024a). Despite these advancements, the intricate mechanism of their multilingual processing behavior remains largely unclear, which leads to an important research question: *How do large language models handle multilingualism?*

To understand the working mechanism of LLMs, existing studies mainly focus on the relationship between model architectures and certain capabilities, with some investigating reasoning abilities with self-attention layers (Hou et al., 2023; Stolfo et al., 2023; Friedman et al., 2023), and others interpreting feed-forward layers as key-value memories for storing factual knowledge (Geva et al., 2021; Dai et al., 2022; Meng et al., 2022). However, these works solely center on English and neglect the multilingual features of LLMs in their interpretations.

---

[†]This work was done during the internship of Yiran Zhao at Alibaba DAMO Academy.

[‡]Wenxuan Zhang is the corresponding author: `isakzhang@gmail.com`

[§]Guizhen Chen is under the Joint Ph.D. Program between DAMO Academy and NTU.

[1]Our code is available at `https://github.com/DAMO-NLP-SG/multilingual_analysis`

38th Conference on Neural Information Processing Systems (NeurIPS 2024).

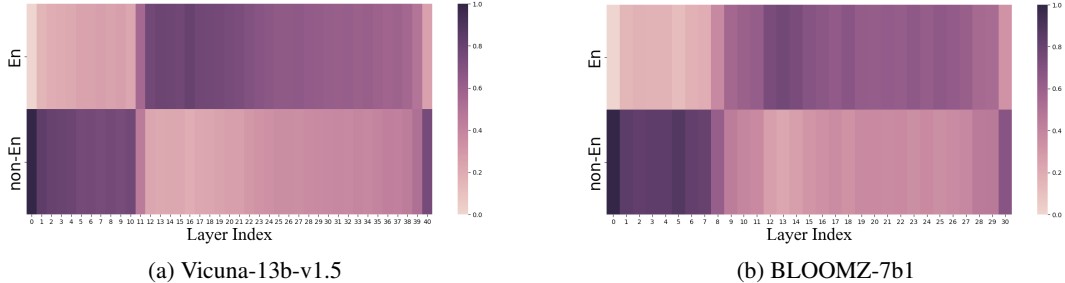

(a) Vicuna-13b-v1.5             (b) BLOOMZ-7b1

Figure 1: Ratio of English and non-English tokens among layers given non-English queries.

To gain an initial understanding of the multilingual mechanism of LLMs, we test LLMs with various non-English queries and decode the hidden embeddings of each layer to tokens within the LLM's vocabulary. Subsequently, we classify these decoded tokens into either English or non-English, and analyze the ratio. Figure 1 illustrates the ratio of English and non-English tokens for each layer of two LLMs. We observe that non-English queries initially generate non-English embeddings as expected. However, as queries progress through the middle layers, the representations surprisingly become English-centric. In the final layers, there is a reversion to predominantly non-English embeddings, matching the non-English queries.

Motivated by the observed transformation above, we hypothesize a three-stage multilingual workflow: *understanding*, *task-solving*, and *generating*. This involves understanding the original non-English queries and interpreting them in English, solving tasks in English, and reverting outputs back to the original language. Furthermore, building upon previous studies that link self-attention structures to reasoning and feed-forward structures to factual knowledge storage (Hou et al., 2023; Geva et al., 2021), we further decouple the task-solving stage into reasoning with self-attention structures and extract-

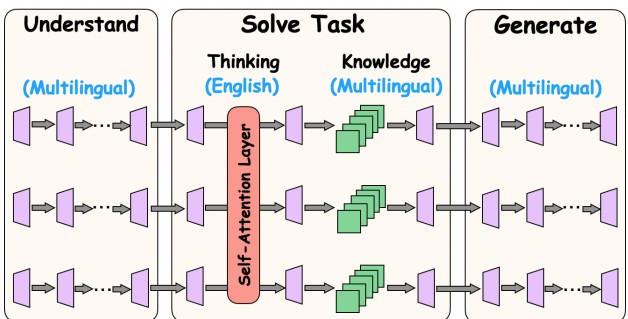

Figure 2: Our hypothesized multilingual workflow, MWork, converts multilingual queries to English for reasoning in English and generates responses in the original language, demonstrating a layered processing approach.

ing multilingual knowledge with feed-forward structures. Therefore, our hypothesized Multilingual Workflow (MWork) illustrated in Figure 2 outlines the three operational stages of LLMs in processing multilingual queries: Initially, LLMs *understand* queries by converting diverse linguistic features into a unified representation. In the *task-solving* phase, LLMs reason in English and incorporate multilingual knowledge to obtain factual content, using self-attention and feed-forward structures, respectively. Finally, models *generate* responses in the original language as the original query.

To verify the proposed MWork, we could extract language-specific parameters, selectively deactivate them within different structures, and observe their corresponding effects, thereby assessing the functionality of corresponding structures and validating our hypothesis. To identify the parameters to be activated, we develop a novel approach called Parallel Language-specific Neuron Detection (PLND). Unlike existing methods that rely on fine-tuning(Frankle and Carbin, 2018; Zhang et al., 2023b), labeled data (Tang et al., 2024; Liu et al., 2024), or parallel corpora (Libovický et al., 2020; Tanti et al., 2021; Zhang et al., 2024) to detect activated parameters, PLND measures the significance of individual neurons with respect to the input in both attention and feed-forward structures without any labeled data or parameter adjustments. Using PLND, we identify language-specific neurons by inputting a free text corpus of that language and isolating consistently activated neurons. We find that by deactivating language-specific neurons which account for only 0.13% of all neurons, LLMs' performance on a multilingual summarization task could drop by 99%.

We then extensively verify the hypothesized MWork framework using the proposed PLND method. Employing various benchmark tasks, including XQuAD (Artetxe et al., 2020) for understanding,

MGSM (Shi et al., 2022) for reasoning, X-CSQA (Lin et al., 2021) for knowledge extraction, and XLSum for generation (Hasan et al., 2021), we selectively deactivate language-specific neurons in each component and verify the functionality of the component by observing a significant decline in performance on the corresponding task. For example, when deactivating the language-specific neurons in the understanding layer, the performance on the multilingual understanding task XQuAD remains stable in English, while experiencing a decrease of $14\%$ in non-English languages. Other tasks exhibit similar pattern when deactivating corresponding neurons. More importantly, with the verified `MWork` framework, enhancing the multilingual capabilities of LLMs can thus be achieved through the fine-tuning of language-specific neurons for certain capabilities. With a remarkable reduction in the training corpus size to a mere few hundred documents, this fine-tuning procedure enhances the multilingual capabilities of LLMs for both high-resource and low-resource languages by an average of $3.6\%$ and $2.3\%$ across all tasks, respectively. Notably, even without an English training corpus, there is a noticeable improvement in English performance, as the enhancement of language-specific neurons yields greater accuracy in enhancing specific languages, while simultaneously ensuring a clear division of parameters among different languages. In summary, the verified `MWork` reveals how LLMs handle multilingual tasks and offers an effective approach for conducting language-specific enhancements without compromising performance in other languages.

## 2    Parallel Language-specific Neuron Detection (`PLND`)

To verify the hypothesized workflow, we propose `PLND` that effectively detects language-specific neurons without relying on any labeled data. In essence, `PLND` identifies neurons crucial for handling individual documents, with language-specific neurons being those that consistently show high importance when processing documents in a particular language.

### 2.1    Sequential Neuron Detection

We define a neuron as a single row or column of a parameter matrix of a language model. To identify neurons responsible for a specific language, it is crucial to discern the significance of a neuron with respect to the inference of a given input. Specifically, when processing the input $c$ in the model, we denote the hidden embedding before the $i$-th layer in Transformer (Vaswani et al., 2017) as $h_i$, and the hidden embedding after the $i$-th layer as $h_{i+1} = T_i(h_i)$, where $T_i$ represents the parameters of the $i$-th layer. For a specific neuron within the $i$-th layer, denoted as $N^{(i)}$, either located in the attention or feed-forward network, we quantify its importance in processing the input $c$ by measuring the difference in the hidden embedding after the $i$-th layer, i.e., $h_{i+1}$, when $N^{(i)}$ is activated or deactivated. Formally, the impact of neuron $N^{(i)}$ for input $c$ is defined as

$$\text{Imp}(N^{(i)}|c) = \|T_i \backslash N^{(i)}(h_i) - T_i(h_i)\|_2, \tag{1}$$

where $T_i \backslash N^{(i)}(\cdot)$ denotes deactivating $N^{(i)}$ in $T_i$, i.e., setting all parameters of the neuron $N^{(i)}$ to zero. With a set of $n$ corpus in a specific language, denoted as $\mathcal{C} = \{c_1, \cdots, c_l, \cdots, c_n\}$, we calculate the importance of each neuron in each layer to each corpus. Furthermore, we can obtain language-specific neurons that are important to all corpus in that language, i.e.,

$$\{N^{(i)} \mid \text{Imp}(N^{(i)}|c_l) \geq \epsilon, \forall c_l \in \mathcal{C}\}, \tag{2}$$

where $\epsilon$ is the pre-defined threshold.

### 2.2    Parallel Neuron Detection

The sequential neuron detection requires traversal of all neurons and inputs sequentially and thus is time-consuming. To address this, we further propose a parallel algorithm for accelerating the process.

**Feed-Forward Network (FFN)**    In the latest open-source models, when processing input $c$, the feed-forward network in a certain layer is defined as

$$\text{FFN}(x) = \Big(\text{SiLU}\big(W_{gate}(x)\big) \cdot W_{up}(x)\Big) W_{down}, \tag{3}$$

where $x \in \mathbb{R}^{l \times d_{model}}$ is the embedding fed into the FFN, $W_{gate}, W_{up} \in \mathbb{R}^{d_{model} \times d_{inter} \, 2}$, $W_{down} \in \mathbb{R}^{d_{inter} \times d_{model}}$. The calculation of the importance of the $k$-th neuron in $W_{up}$, when processing the input $c$, as presented in Equation 1, can be equivalently transformed to

$$\text{Imp}(W_{up}[:,k]|c) = \|\hat{\text{FFN}}(x) - \text{FFN}(x)\|_2 = \left\| \big(h_{\text{ffn}} \cdot \text{Mask}[k]\big) W_{down}(x) \right\|_2, \tag{4}$$

where $h_{\text{ffn}} \in d_{inter}$ represents the embedding before $W_{down}$, and $\text{Mask}[k] \in d_{inter}$ is a vector with the $k$-th element equal to $1$ and the rest equal to $0$. To calculate $\text{Imp}(W_{up}[:,k]|c)$ for $k \in d_{inter}$ parallelly, we introduce a diagonal mask matrix of size $(d_{inter}, d_{inter})$, denoted as $\text{Mask}$. Therefore,

$$\text{Imp}(W_{up}|c) = \|(h_{\text{ffn}} \cdot \text{Mask}) W_{down}(x)\|_2. \tag{5}$$

Furthermore, we observe that deactivating the $k$-th neuron of $W_{down}$ is equivalent to deactivating the $k$-th neuron in $W_{up}$, as they both result in $h_{\text{ffn}}[k] = 0$. Hence, we can also derive $\text{Imp}(W_{down}|c)$ by employing Equation (5).

**Self-Attention Network**   When processing input $c$, the self-attention network in a certain layer is

$$\text{Attention}(x) = \text{Softmax}\Big(\frac{W_Q(x)W_K^T(x)}{\sqrt{d}}\Big)W_V(x), \tag{6}$$

where $W_Q, W_K, W_V \in \mathbb{R}^{d_{model} \times d_{mid}}$. [3] Since $W_V(x)$ is not in the non-linear softmax calculation, we can calculate $\text{Imp}(W_V|c)$ by applying Equation (5). For $W_Q$, we obtain $\text{Imp}(W_Q[:,k]|c)$ by deactivating its $k$-th neuron, specifically, $\hat{W}_Q \leftarrow W_Q[:,k] = 0$. Firstly, we calculate the difference in attention weight before and after deactivation, prior to scaling and softmax,

$$\Delta_k(x) = W_Q(x)W_K^T(x) - \hat{W}_Q(x)W_K^T(x) = W_Q(x)[:,k]W_K(x)[k,:] \in \mathbb{R}^{l \times l}. \tag{7}$$

Next, as the changes in attention exhibit a positive correlation with the changes in the output of this layer, the importance of $W_Q[:,k]$ in processing $c$, as defined in Equation 1, can be approximated as

$$\begin{aligned}
\text{Imp}(W_Q[:,k]|c) &\approx \|\hat{\text{attention}}(x) - \text{attention}(x)\|_2 \\
&\approx \left\| \text{softmax}\Big(\frac{W_Q(x)W_K^T(x) - \Delta_k(x)}{\sqrt{d}}\Big) - \text{softmax}\Big(\frac{W_Q(x)W_K^T(x)}{\sqrt{d}}\Big) \right\|_2.
\end{aligned} \tag{8}$$

This process can also be calculated in parallel, specifically,

$$\begin{aligned}
\Delta(x) &= W_Q(x)W_K^T(x) - \hat{W}_Q(x)W_K^T(x) \\
&= W_Q(x).resize(l,1,d_{mid}) \times W_K(x).resize(1,l,d_{mid}) \in \mathbb{R}^{l \times l \times d_{mid}}.
\end{aligned} \tag{9}$$

Therefore, the importance of $W_Q$ in processing input $c$ is calculated by

$$\text{Imp}(W_Q|c) \approx \left\| \text{softmax}\Big(\frac{W_Q(x)W_K^T(x) - \Delta(x)}{\sqrt{d}}\Big) - \text{softmax}\Big(\frac{W_Q(x)W_K^T(x)}{\sqrt{d}}\Big) \right\|_2. \tag{10}$$

Similarly, since $W_K$ is symmetrical to $W_Q$, $\text{Imp}(W_K|c)$ can be calculated in the same way.

### 2.3   Detection of Language-Specific Neurons

We then apply PLND to selected languages and models to validate its effectiveness in detecting language-specific neurons and to further investigate the relationships between languages.

**Experimental Setup.**   We test two open-source models that perform well on multilingual tasks, including *Vicuna-7b-v1.5*[4] (Chiang et al., 2023) and *Mistral-7b-Instruct-v0.2* (Jiang et al., 2023). For simplicity, we abbreviate them as Vicuna and Mistral hereafter to represent the two models respectively. We select the text summarization task with the XLSum (Hasan et al., 2021) dataset as the reference task to evaluate multilingual performance as it requires the model to comprehend the

---

[2]$W(\cdot)$ represents the linear matrix product of the input $x$ and the parameter $W$, i.e., $W(x) := xW$.

[3]In some models like Vicuna and Mistral, $d_{model} = d_{mid}$, but we use different notations to avoid ambiguity.

[4]We do not directly utilize Llama2-chat as it does not follow multilingual instructions, consistently responding in English regardless of the language of the query.

Table 1: Multilingual performance on XLSum when deactivating language-specific neurons ("Lang-Spec") and an equivalent number of randomly selected neurons ("Random").

| Model | Method | Fr | Zh | Es | Ru | Avg. |
|---|---|---|---|---|---|---|
| **Vicuna** | Original | 14.2 | 61.1 | 10.4 | 20.8 | 26.6 |
| | Deactivate Random | 14.1 | 61.6 | 10.4 | 20.8 | 26.7 |
| | Deactivate Lang-Spec | **0.83** | **0.00** | **0.24** | **0.42** | **0.37** |
| **Mistral** | Original | 15.2 | 56.4 | 10.6 | 21.0 | 25.8 |
| | Deactivate Random | 15.4 | 55.9 | 10.2 | 21.2 | 25.7 |
| | Deactivate Lang-Spec | **0.21** | **0.39** | **0.15** | **0.07** | **0.21** |

input text and generate a coherent fragment. We adopt 4 high-resource languages including French (Fr), Chinese (Zh), Spanish (Es), and Russian (Ru), as their initial performance on those languages is already quite reasonable for observing the multilingual processing mechanism. Furthermore, we utilize OSCAR (Caswell et al., 2020) corpus which contains web crawling texts for each language to compile a language-specific corpus without task-specific considerations. More details are presented in Appendix B.

**Existence of Language-Specific Neurons**    Using `PLND`, we feed a corpus in a specific language to LLMs and identify neurons that are consistently activated, which are responsible for processing queries in that language. To ascertain whether these neurons are genuinely language-specific, we assess the performance of LLMs in corresponding languages when these neurons are deactivated versus when the same number of randomly sampled neurons are deactivated.

Table 1 demonstrates the decline of multilingual capabilities when deactivating language-specific neurons. Although just deactivating around $0.13\%$ neurons, LLMs lose their multilingual capabilities and fail to generate meaningful content. In contrast, deactivating the same number of randomly selected neurons does not yield any difference. Therefore, the detected neurons are language-specific and related to handling corresponding multilingual inputs.

## 2.4    Analysis of Language-Specific Neurons

We further investigate the degree of overlap among their language-specific neurons. Our findings reveal that in both Mistral and Vicuna, English shows limited overlap with other languages, indicating many language-specific neurons, while languages within the same family, such as Spanish, French, and English, demonstrate more overlap. More details are illustrated in Appendix C.

In addition, we examine two more types of multilingual LLMs, including BLOOMZ (Muennighoff et al., 2023), a *hyper-multilingual* LLM claiming to support 46 languages, and Chinese Llama (Cui et al., 2023), a *bilingual* LLM focusing on English and Chinese. We find that language-specific neurons in BLOOMZ follow patterns similar to Mistral and Vicuna. However, in Chinese LLama, Chinese dominates as the primary language for reasoning and knowledge extraction across all languages, with notably absent language-specific neurons. Details are shown in Appendix D.

Given the certain overlap ratio of language-specific neurons from other languages with those of English, as illustrated in the first column of Figure 5 and Figure 6, we conduct supplementary experiments to demonstrate that these neurons are not language-agnostic neurons crucial for general comprehension and logical reasoning (Liang et al., 2024; Tang et al., 2024). Instead, these overlapping neurons represent only a subset of language-specific neurons, while the language-agnostic neurons responsible for essential understanding and reasoning are those not identified as language-specific. Further elaboration and detailed results are presented in Appendix E.

# 3    Multilingual Workflow (`MWork`) of LLMs

## 3.1    `MWork`

By classifying the hidden representations of each layer in LLMs into English or non-English (as shown in Figure 1), we can observe the shift from non-English to English-centric, and back to non-English with the progression through the layers. This motivates us to hypothesize a three-stage multilingual workflow: *understanding* the original non-English queries and interpreting them in

English, *task-solving* in English, and *generating* back to the original language. Nevertheless, the presence of certain non-English tokens during the English-centric task-solving stage inspires us to further investigate this stage.

With the proposed `PLND` method, we extract language-specific neurons from attention and feed-forward structures when processing various multilingual queries. We plot the average number of activated language-specific neurons of Mistral when processing each query in Figure 3. Notably, the number of language-specific neurons decreases within the self-attention structure in the task-solving layer but remains consistent across the layers of the feed-forward structure. This decline implies a reliance on the English language for reasoning while extracting multilingual knowledge to support query processing, which is also consistent with (Geva et al., 2021)'s interpretation of the feed-forward structure as key-value memories for knowledge extraction. Therefore, we further decompose the task-solving layer into two parts: *reasoning* in English and *extracting knowledge* in a multilingual context.

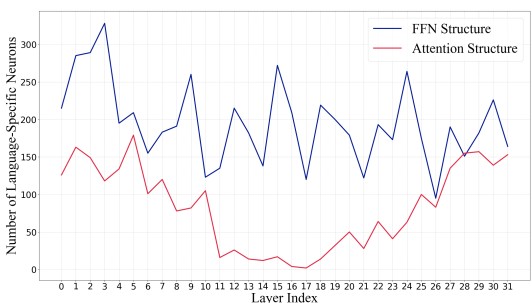

Figure 3: Number of language-specific neurons when processing multilingual queries.

Considering the above insights, we propose the `MWork` hypothesis for explaining LLM's multilingual workflow: LLMs first *understand* user input by unifying diverse linguistic features. They then engage in the *task-solving* phase, employing English for reasoning and leveraging multilingual knowledge through self-attention and feed-forward structures, respectively. Finally, the models *generate* responses aligned with the query's original language.

## 3.2 Verification Experiment Setup

To verify `MWork`, we selectively deactivate language-specific neurons from each component. Then its functionality can be verified if this deactivation results in minimal impact on English performance while exhibiting a notable decline in multilingual performance for the corresponding task.

**Dataset**   To comprehensively understand how LLMs work with different abilities, we employ four kinds of tasks including MGSM (Shi et al., 2022) for reasoning task, XQuAD (Artetxe et al., 2020) for understanding task, X-CSQA (Lin et al., 2021) for knowledge question answering task, and XLSum (Hasan et al., 2021) for generation task. Detailed information regarding these datasets and the testing prompts can be found in Appendix F. We adopt 6 languages including English (En), German (De), French (Fr), Chinese (Zh), Spanish (Es), and Russian (Ru), as their initial performance on those languages is already quite reasonable for observing the multilingual processing mechanism. For XLSum, we randomly sample 500 data points from the whole test set for each language taking into consideration its long inference time, while for other tasks, we employ the entire test set. We evaluate the vanilla performance of Vicuna and Mistral on these datasets for later comparison as presented in Appendix G. For reasoning, understanding, and knowledge question answering tasks, we adopt accuracy as the metric. As for the generation tasks, we adopt ROUGE-L as the metric.

**Deactivation Strategy**   We primarily consider two aspects when selecting the deactivation settings: (1) language-specific neurons versus randomly chosen neurons, and (2) the position of neurons, which encompasses four structures. Note that for a fair comparison, we ensure the numbers of deactivated neurons in all settings are the same. More detailed settings are explained from Section 3.3 to Section 3.6. For the concrete numbers of different layers, we tune hyperparameters by XQuAD in Chinese. Details are explained in Appendix H.

**Notations**   Tables 2 to 5 present the results of deactivating certain neurons, where "Under" denotes the understanding layers, "S-ATTN" and "S-FFN" correspond to the self-attention and the feed-forward structures within the task-solving layers respectively, "Gen" refers to the generation layers. The term "Random" is used to describe deactivating randomly chosen neurons, whereas "Lang-Spec" refers to the deactivation of language-specific neurons. We also present the gap between the original performance (as shown in Table 11) and performance after deactivation (as shown in Table 14 to Table

Table 2: Results of the **understanding** task, where '✗' indicates that chosen neurons in the corresponding layer are deactivated, and '✓' signifies they are activated. $\Delta$ is defined as the difference between the reduction in performance in English, denoted as $\Delta_{\text{Eng}}$, and the reduction in performance in non-English languages, denoted as $\Delta_{\text{n-Eng}}$.

| Model | Deactivating Method | | | | | Performance | | | | |
| | Under | S-ATTN | S-FFN | Gen | Neuron | Eng | n-Eng | $\Delta_{\text{Eng}}$ | $\Delta_{\text{n-Eng}}$ | $\Delta\uparrow$ |
|---|---|---|---|---|---|---|---|---|---|---|
| Vicuna | ✗ | ✓ | ✓ | ✓ | Random | 57.8 | 53.9 | +0.3 | −0.1 | +0.4 |
| | ✗ | ✗ | ✗ | ✗ | Random | 57.9 | 54.2 | +0.4 | +0.3 | +0.1 |
| | ✓ | ✗ | ✗ | ✓ | Lang-Spec | 40.9 | 38.6 | −15.9 | −15.3 | −0.6 |
| | ✓ | ✓ | ✓ | ✗ | Lang-Spec | 57.9 | 52.8 | −0.4 | −1.1 | +0.7 |
| | ✗ | ✓ | ✓ | ✓ | Lang-Spec | 56.5 | 46.0 | −0.5 | −7.9 | +7.4 |
| Mistral | ✗ | ✓ | ✓ | ✓ | Random | 58.1 | 55.5 | +1.0 | −0.2 | +1.2 |
| | ✗ | ✗ | ✗ | ✗ | Random | 57.6 | 55.5 | +0.5 | −0.2 | +0.7 |
| | ✓ | ✗ | ✗ | ✓ | Lang-Spec | 53.2 | 47.0 | −3.9 | −8.7 | +4.8 |
| | ✓ | ✓ | ✓ | ✗ | Lang-Spec | 56.4 | 54.6 | −0.7 | −1.0 | +0.3 |
| | ✗ | ✓ | ✓ | ✓ | Lang-Spec | 56.2 | 48.3 | −0.9 | −7.4 | +6.5 |

17) for English ($\Delta_{\text{Eng}}$) and averaged non-English languages ($\Delta_{\text{n-Eng}}$), respectively. A single metric $\Delta$ is then introduced as $\Delta_{\text{Eng}} - \Delta_{\text{n-Eng}}$, where a high value indicates such deactivation operation does not bring much impact to the English performance but lead to performance drop in non-English. Therefore, this provides a direct single indicator that the deactivated neurons are language-specific and hold a significant responsibility in executing the corresponding task.

## 3.3 Verify the Understanding Stage in `MWork`

**Deactivating Method**    Table 2 shows the results of the understanding task following the deactivation of five distinct sets of neurons: (i) neurons randomly selected from the understanding layers; (ii) neurons randomly chosen across all layers; (iii) language-specific neurons within the task-solving layers; (iv) language-specific neurons in the generation layers; (v) language-specific neurons in the understanding layers. As mentioned above, in order to verify the functionality of the understanding layer (setting v), we compare it with deactivating other types of layers, specifically setting iii for the task-solving layer and setting iv for the generation layer. Full results are listed in Appendix I.

**Findings**    We find that by deactivating randomly sampled neurons, no matter in the understanding layer or all layers, the performance of LLMs in both English and non-English languages is almost unaffected compared to other deactivating methods. Note that in some cases, deactivating randomly sampled neurons may even increase the performance because irrelevant neurons are removed, which also aligns with the finding from (Sharma et al., 2023). When assessing the differential impact on English and non-English language performance after the deactivation, specifically the difference calculated as $\Delta_{\text{Eng}} - \Delta_{\text{n-Eng}}$, it is evident that the deactivation of random neurons within the understanding layer amplifies this effect. This observation lends partial support to the hypothesized role of the understanding layer in language processing.

Furthermore, we find that deactivating language-specific neurons in the understanding layer influences the performance in English a little while significantly decreasing the performance in non-English languages. When deactivating language-specific neurons in the task-solving layer, both English and non-English languages are significantly reduced while deactivating language-specific neurons in the generation layer influences a little for both English and non-English languages. Therefore, we prove that the first several layers are responsible for understanding because deactivated neurons just disable LLMs on the NLU task in non-English languages. Furthermore, disabling language-specific neurons in the task-solving layer shows that LLMs rely on English, as performance drops across all languages.

## 3.4 Verify the Reasoning Structure in `MWork`

**Deactivating Method**    Table 3 shows the result of the reasoning task, where we deactivate 6 sets of neurons. We adhere to the previous logic of selecting deactivation settings, with the exception that

Table 3: Results of the **reasoning** task. Disabling all language-specific neurons, except for those involved in self-attention structure within the task-solving layer, greatly reduces performance.

| Model | Deactivating Method | | | | | Performance | | | | |
|---|---|---|---|---|---|---|---|---|---|---|
| | Under | S-ATTN | S-FFN | Gen | Neuron | Eng | n-Eng | $\Delta_{Eng}$ | $\Delta_{n\text{-}Eng}$ | $\Delta\uparrow$ |
| Vicuna | ✓ | ✗ | ✓ | ✓ | Random | 20.0 | 11.3 | −0.4 | −1.8 | +1.4 |
| | ✓ | ✗ | ✗ | ✓ | Random | 18.4 | 12.2 | −2.0 | −1.0 | −1.0 |
| | ✗ | ✗ | ✗ | ✗ | Random | 19.6 | 12.5 | −0.8 | −0.7 | −0.1 |
| | ✓ | ✗ | ✗ | ✓ | Lang-Spec | 7.2 | 3.4 | −13.2 | −9.8 | −3.4 |
| | ✗ | ✓ | ✓ | ✗ | Lang-Spec | 18.1 | 8.3 | −2.3 | −4.9 | +2.6 |
| | ✗ | ✓ | ✗ | ✗ | Lang-Spec | 19.0 | 7.8 | −1.4 | −5.4 | +4.0 |
| Mistral | ✓ | ✗ | ✓ | ✓ | Random | 40.8 | 23.4 | −5.2 | −2.9 | −2.3 |
| | ✓ | ✗ | ✗ | ✓ | Random | 39.2 | 24.0 | −6.8 | −2.3 | −4.5 |
| | ✗ | ✗ | ✗ | ✗ | Random | 45.2 | 26.8 | −0.8 | +0.5 | −1.3 |
| | ✓ | ✗ | ✗ | ✓ | Lang-Spec | 38.2 | 18.4 | −7.8 | −7.9 | +0.1 |
| | ✗ | ✓ | ✓ | ✗ | Lang-Spec | 44.0 | 18.1 | −2.0 | −8.2 | +6.2 |
| | ✗ | ✓ | ✗ | ✗ | Lang-Spec | 46.2 | 18.3 | +0.2 | −8.0 | +8.2 |

Table 4: Results of the **knowledge** question answering task. The highest performance reduction difference ($\Delta$) is achieved by disabling all language-specific neurons in the feed-forward structure within the task-solving layer.

| Model | Deactivating Method | | | | | Performance | | | | |
|---|---|---|---|---|---|---|---|---|---|---|
| | Under | S-ATTN | S-FFN | Gen | Neuron | Eng | n-Eng | $\Delta_{Eng}$ | $\Delta_{n\text{-}Eng}$ | $\Delta\uparrow$ |
| Vicuna | ✓ | ✓ | ✗ | ✓ | Random | 57.5 | 39.5 | −0.3 | +0.0 | −0.3 |
| | ✓ | ✗ | ✗ | ✓ | Random | 56.0 | 38.7 | −1.8 | −0.8 | −1.0 |
| | ✗ | ✗ | ✗ | ✗ | Random | 57.7 | 39.6 | −0.1 | +0.1 | −0.2 |
| | ✓ | ✗ | ✓ | ✓ | Lang-Spec | 33.7 | 30.3 | −24.1 | −9.2 | −14.9 |
| | ✓ | ✓ | ✗ | ✓ | Lang-Spec | 57.5 | 37.5 | −0.3 | −2.0 | +1.7 |
| Mistral | ✓ | ✓ | ✗ | ✓ | Random | 61.0 | 37.0 | −0.3 | −0.5 | +0.2 |
| | ✓ | ✗ | ✗ | ✓ | Random | 60.7 | 36.3 | −0.6 | −1.2 | +0.6 |
| | ✗ | ✗ | ✗ | ✗ | Random | 61.8 | 37.4 | +0.1 | −0.1 | +0.2 |
| | ✓ | ✗ | ✓ | ✓ | Lang-Spec | 51.2 | 28.9 | −10.1 | −8.6 | −1.5 |
| | ✓ | ✓ | ✗ | ✓ | Lang-Spec | 61.2 | 35.1 | −0.1 | −2.4 | +2.3 |

we do not conduct an independent experiment on deactivating neurons in the understanding layer, as its functionality has already been verified. Details are listed in Appendix I.

**Findings** We find that deactivating randomly sampled neurons in task-solving layers disables the capabilities of LLMs in reasoning to a greater extent than deactivating randomly sampled neurons in all layers, which verifies the function of the task-solving layer. Furthermore, comparing three deactivating language-specific neuron methods, we find that deactivating the task-solving layer decreases performance in both English and non-English. On the contrary, when we only deactivate language-specific neurons not in the task-solving layer, non-English is influenced more seriously than English. Moreover, eliminating interference from the feed-forward layer achieves better results, which verifies the function of attention structure in the task-solving layer.

### 3.5 Verify the Knowledge Extraction Structure in `MWork`

**Deactivating Method** Table 4 shows the result of the knowledge question answering task, where we deactivate 5 sets of neurons. Similarly, we exclude the deactivation of neurons in layers that have already been verified and instead concentrate on the self-attention structure and feed-forward structure in the task-solving layer. Details are listed in Appendix I.

**Findings** Likewise, targeted deactivation of language-specific neurons within the feed-forward structure of the task-solving layer predominantly affects non-English languages. This implies that

Table 5: Results of the **generation** task. The highest performance reduction difference ($\Delta$) is achieved by disabling all language-specific neurons in the generation layer.

| Model | Deactivating Method | | | | | Performance | | | | |
|---|---|---|---|---|---|---|---|---|---|---|
| | Under | S-ATTN | S-FFN | Gen | Neuron | Eng | n-Eng | $\Delta_{\text{Eng}}$ | $\Delta_{\text{n-Eng}}$ | $\Delta \uparrow$ |
| Vicuna | ✓ | ✓ | ✓ | ✗ | Random | 13.2 | 26.8 | +0.1 | +0.1 | +0.0 |
| | ✗ | ✗ | ✗ | ✗ | Random | 13.0 | 26.7 | −0.1 | +0.0 | −0.1 |
| | ✓ | ✓ | ✓ | ✗ | Lang-Spec | 13.1 | 25.7 | +0.0 | −1.1 | +1.1 |
| Mistral | ✓ | ✓ | ✓ | ✗ | Random | 13.6 | 25.9 | +0.1 | +0.1 | +0.0 |
| | ✗ | ✗ | ✗ | ✗ | Random | 13.6 | 25.7 | +0.1 | −0.2 | +0.3 |
| | ✓ | ✓ | ✓ | ✗ | Lang-Spec | 13.8 | 24.3 | +0.3 | −1.5 | +1.8 |

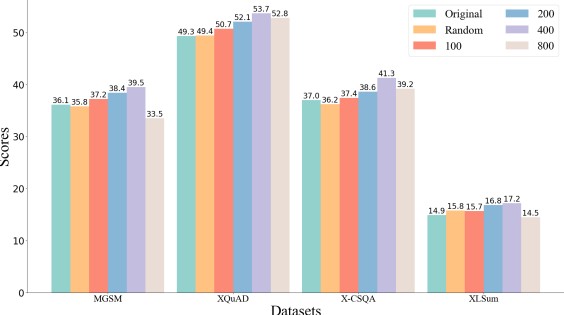

Figure 4: Enhancement results on high-resource languages, while the number is average among languages.

Table 6: Enhancement is achieved by fine-tuning Mistral-7b-v0.1 model utilizing 400 documents from each language correspondingly. The results are averaged across four tasks. Performance on English ("En") is obtained by averaging the results from four fine-tuned models.

| Method | En | Vi | Th | Ar | Sw |
|---|---|---|---|---|---|
| Original | 41.1 | 32.7 | 25.6 | 21.7 | 15.1 |
| Random | 40.8 | 32.7 | 25.2 | 21.2 | 15.1 |
| Lang-Spec | **44.6** | **34.9** | **28.5** | **23.4** | **16.9** |

processing multilingual queries necessitates accessing the multilingual information embedded within the relevant structures. However, disabling the self-attention structure compromises the ability to solve tasks across all languages.

### 3.6 Verify the Generation Structure in `MWork`

**Deactivating Method**    Table 5 shows the result of the generation task, where we deactivate 3 sets of neurons. Since all previous layers have been verified, we solely deactivate neurons in the generation layer and compare them with randomly selected neurons. Details are listed in Appendix I.

**Findings**    Similar to other tasks, the disabling of language-specific neurons within the generation layer diminishes their capacity to generate content in the respective languages. By selectively deactivating neurons that are not associated with English, we do not completely eliminate the models' multilingual generation abilities. However, as demonstrated in Table 1, the complete deactivation of all language-specific neurons results in the total loss of the LLMs' multilingual generation capabilities.

## 4   Multilingual Enhancement with `MWork`

We have verified `MWork` for explaining the multilingual working mechanism of LLMs in the above section via deactivating certain neurons. While opposite to employing deactivation, we can also enhance their multilingual ability, especially the understanding and generating ability, by fine-tuning these language-specific neurons. With language-specific neurons comprising only around 0.1% of all parameters, the need for training documents to improve multilingual capabilities can be significantly reduced to just a few hundred. Additionally, fine-tuning only the language-specific neurons for a particular language does not impact performance in other languages, allowing us to enhance specific languages while preserving performance in others.

**`MWork` helps with enhancing multilingual ability by hundreds of documents.**    We employ *Mistral-7b-v0.1* for enhancement to eliminate the interference of instruction fine-tuning, and select causal language modeling as our training task. We create a dataset comprising $\{100, 200, 400, 800\}$

randomly selected documents for each language, extracted from the Wikipedia corpus (Foundation). Figure 4 shows the results of enhancement on high-resource languages (De, Fr, Zh, Es, Ru). The numbers represent the sizes of the training corpus when fine-tuning language-specific neurons, while "Random" represents the fine-tuning of an equivalent number of randomly chosen neurons using a corpus of $400$. Our findings reveal that fine-tuning with a few hundred documents yields significant performance improvements on multilingual tasks: $3.4\%$ on MGSM, $4.4\%$ on XQuAD, $4.3\%$ on X-CSQA, and $2.3\%$ on XLSum. Moreover, English performance is enhanced by an average of $3.7\%$ across all tasks. These results further confirm the effectiveness of `MWork` in interpreting structure functionality for LLM's multilingual query handling, offering precise and independent methods for multilingual enhancement. When fine-tuning with 800 documents, the performance deteriorates compared to using 400 documents. This drop can be attributed to the incorporation of additional knowledge, which disrupts the original knowledge distribution and leads to overfitting of the model to Wikipedia. This can be addressed by mixing data from more sources such as textbooks or websites.

In addition, we verify the effectiveness of such enhancement method on low-resource languages, given that low-resource performance is relatively low with the original model. We select four languages including Vietnamese (Vi), Thai (Th), Arabic (Ar), and Swahili (Sw), covering languages with both latin and non-latin scripts and having corresponding testing set in our considered benchmarks. The model was then evaluated on four benchmarks, and the result shown in Table 6 is the average scores among tasks. It is evident that the fine-tuning method using language-specific neurons enhances the model's multilingual performance in low-resource languages by an average of $2.2\%$. Notably, the improvement of $3.5\%$ in English performance is observed even without an English training corpus, indicating the effectiveness of the distinct language responsibilities assigned to neurons.

## 5 Related Work

In the era of LLMs, numerous studies have been conducted to develop multilingual benchmarks (Zhang et al., 2023a), enhance multilingual performance without parameter adjustments through translation (Liang et al., 2023; Huang et al., 2023), aligning representations (Nguyen et al., 2023a; Salesky et al., 2023), prompting (Li et al., 2023b; Tanwar et al., 2023). Furthermore, certain works focus on improving multilingual abilities for a single task via cross-lingual transfer (Kim et al., 2017; Lin et al., 2019; Pfeiffer et al., 2020; Zhao et al., 2024b), while others aim to enhance multilingual proficiency by continuous training in one language to obtain mono-lingual LLMs (Cui et al., 2023), or in multiple domain languages to obtain domain-lingual LLMs (Nguyen et al., 2023b). Additionally, some works achieve multilingual LLMs by training from scratch (Muennighoff et al., 2023). However, these studies are limited to specific task types or require substantial training corpora due to a lack of comprehensive understanding of the multilingual mechanisms of LLMs.

Conventional interpretability research investigates the significance of input features with their corresponding outputs (Vig, 2019; Hewitt and Liang, 2019; Qiu et al., 2020). In the era of LLMs, one brunch of work includes efforts to understand knowledge storage, with (Geva et al., 2021) initiating the study of the feed-forward layer as a knowledge base. Subsequent work has furthered this by altering neuron values (Dai et al., 2022), mapping embeddings to words (Geva et al., 2022), modifying inputs to recover embeddings (Meng et al., 2022), and analyzing attention heads (Li et al., 2023a). Another line of research centers on the self-attention layer, examining its connection to reasoning capability (Hou et al., 2023; Stolfo et al., 2023; Friedman et al., 2023) by contrasting the reasoning tree based on attention weights.

## 6 Conclusion

In this work, we examine how LLMs handle multilingualism. The proposed multilingual workflow (`MWork`) suggests that LLMs initially understand queries by converting multilingual inputs into English, reason in English in intermediate layers while incorporating multilingual knowledge, and generate responses aligned with the original language in the final layers. The validity of `MWork` is verified using Parallel Language-specific Neuron Detection (`PLND`), which identifies activated neurons for different languages without labeled data. By detecting language-specific neurons and fine-tuning them with a small training corpus, `MWork` enhances multilingual abilities in specific languages without compromising others, resulting in significant improvements across tasks.

## Acknowledgement

This work was substantially supported by DAMO Academy through DAMO Academy Research Intern Program. This research is partially supported by the National Research Foundation Singapore under the AI Singapore Programme (AISG Award No: AISG2-TC-2023-010-SGIL) and the Singapore Ministry of Education Academic Research Fund Tier 1 (Award No: T1 251RES2207).

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

## A    English and Non-English Tokens

We employ `cld3` package to detect the language of each token in the embeddings of each layer, which is a language detection library based on the Compact Language Detector 3 model developed by Google. Furthermore, if the detection result is reliable, i.e., cld3.get_language(token).is_reliable $== True$, we adopt the detection results, otherwise the token is categorized as a non-word.

## B    Multilingual Corpus

Note that our selection criterion for the number of documents is based on achieving substantial coverage of each language's vocabulary, ensuring that the selected contexts provide a representative sample of the language, as shown in Table 7.

Table 7: Corpus details across languages are tailored to encompass the majority of each language's vocabulary, where "corpus size" indicates the number of contexts selected, "corpus vocab" represents the vocabulary coverage within the selected contexts, "vocab size" refers to the number of vocabularies of that language.

| Language | En | De | Fr | Zh | Es | Ru |
|---|---|---|---|---|---|---|
| Corpus Size | 180k | 30k | 50k | 20k | 20k | 20k |
| Corpus Vocab | 249k | 154k | 134k | 198k | 90k | 144k |
| Vocab Size | 273k | 148k | 135k | 329k | 93k | 150k |

## C    Interrelation of Language-Specific Neurons Across Languages

Using neurons identified by `PLND`, we investigate the relationships between languages via the degree of overlap among their language-specific neurons, defined as

$$\text{overlap}(x, y) = \frac{|\mathcal{N}_x \cap \mathcal{N}_y|}{|\mathcal{N}_y|}, \tag{11}$$

where $\mathcal{N}_{language}$ represents the set of detected language-specific neurons. Figure 5 shows the neuron overlapping ratio overlap$(x, y)$ of any two languages in different structures of two models.

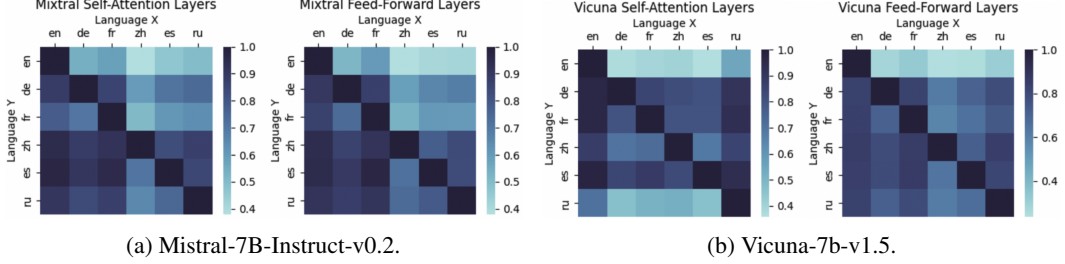

(a) Mistral-7B-Instruct-v0.2.                    (b) Vicuna-7b-v1.5.

Figure 5: Overlapping ratio of language-specific neurons in self-attention and feed-forward structures.

We can observe that in both Mistral and Vicuna, the intersection with English from other languages is relatively limited (i.e., the first row of each figure), suggesting that English possesses a predominant number of language-specific neurons. Additionally, there is a pronounced tendency for languages belonging to the same family to demonstrate a higher degree of overlap with each other, such as Spanish, French, and English.

## D    Analysis on Different Multilingual LLMs

We further examine two more types of multilingual LLMs, including BLOOMZ (Muennighoff et al., 2023), a *hyper-multilingual* LLM claiming to support 46 languages, and Chinese Llama (Cui et al., 2023), a *bilingual* LLM focusing on English and Chinese.

**Hyper-Multilingual LLMs** Figure 6 illustrates the degree of neuron overlap among languages within both the self-attention and feed-forward structures of BLOOMZ. In contrast to the findings shown in Figure 5, there is a marked reduction in overlap, indicating that individual languages maintain a higher degree of independence and do not extensively share neurons with one another.

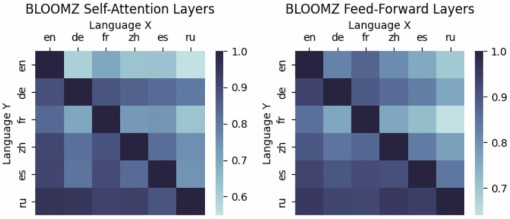

Figure 6: Overlapping ratio of language-specific neurons in BLOOMZ

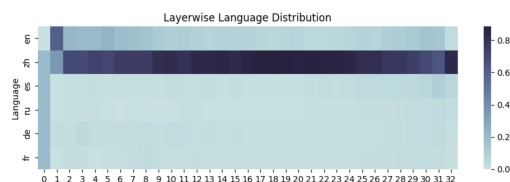

Figure 7: Ratio of languages among layers in Chinese Llama given non-English instructions.

**Bilingual LLMs** We employ Chinese Llama (Cui et al., 2023), which extends existing vocabulary and incorporate secondary pre-training using Chinese data and fine-tune the model with Chinese instruction datasets. However, this intensive training can lead to a degradation in performance for languages other than Chinese. As depicted in Figure 7, Chinese predominates as the primary language for reasoning processing and knowledge extraction across all languages. Consequently, the absence of language-specific neurons results in the transformation of it into a Chinese-centric LLM.

# E Language-Agnostic Neurons

We initially implement a radical deactivation approach, wherein we specifically deactivate overlapping elements between each language and English. These elements precisely correspond to the intersecting neurons in the first column of Figure 5. Presented below are the comprehensive findings pertaining to Mistral. Our evaluation is centered around the reasoning task, which is recognized as the most indicative and challenging assessment for the model. We compare under the optimal "deactivating" method, which involves deactivating all language-specific neurons except those in S-ATTN.

Table 8: Performance of deactivating language-specific neurons without overlapped between English.

| Language | Eng | non-Eng | $\Delta_{\text{Eng}}$ | $\Delta_{\text{non-Eng}}$ | $\Delta \uparrow$ |
|---|---|---|---|---|---|
| All language-specific neurons | 46.2 | 18.3 | +0.2 | −8.0 | +8.2 |
| LSN without overlapped between English | 45.8 | 20.2 | −0.2 | −6.1 | +5.9 |

As evident by Table 8, the performance of English remains stable, contrasting sharply with the significant decline in the performance of multilingual. Removing overlapped neurons, as opposed to deactivating all language-specific neurons, leads to a less pronounced drop, yet the impact remains noteworthy. This demonstrates that overlapped neurons are not language-agnostic; they are not utilized for general comprehension and logical reasoning. Otherwise, the fundamental reasoning capacity and performance in multilingual contexts would remain unaffected. In addition, we retained the language-specific neurons that overlapped in all languages, meaning that we removed them from the language-specific neurons to be deactivated. Detailed results follow.

Table 9: Performance of deactivating language-specific neurons without all languages overlapped.

| Language | Eng | non-Eng | $\Delta_{\text{Eng}}$ | $\Delta_{\text{non-Eng}}$ | $\Delta \uparrow$ |
|---|---|---|---|---|---|
| All language-specific neurons | 46.2 | 18.3 | +0.2 | −8.0 | +8.2 |
| LSN without all languages overlapped | 45.6 | 18.7 | −0.4 | −7.6 | +7.2 |

The neurons that overlap across all languages only account for $0.02\%$ of the total number of neurons. From the results in Table 9, we can see that the performance is almost the same as deactivating all language-specific neurons. This further proves that these neurons are not language-agnostic neurons, but only a subset of language-specific neurons.

Table 10: Zero-shot prompts for each dataset.

| Task | Zero-Shot Prompt |
|------|------------------|
| MGSM | Let's think step by step. Question: {question} |
| XQuAD | {context} Question: {question} |
| XLSum | Summarize the context in one sentence. Title: {title} Context: {article} |
| X-CSQA | Question: {question} |

Table 11: Assessing the baseline performance of Vicuna and Mistral across four representative multilingual tasks in selected languages, where Avg. is calculated among non-English languages.

| Model | Task | En | De | Fr | Zh | Es | Ru | Avg. |
|-------|------|-----|-----|-----|-----|-----|-----|------|
| **Vicuna** | XQuAD | 57.5 | 50.3 | — | 55.7 | 55.7 | — | 53.9 |
| | MGSM | 20.4 | 14.8 | 14.8 | 12.8 | 13.2 | 10.0 | 13.1 |
| | X-CSQA | 57.8 | 43.8 | 40.1 | 43.2 | 44.3 | 26.0 | 39.5 |
| | XLSum | 13.1 | — | 14.2 | 61.1 | 10.4 | 20.8 | 26.6 |
| **Mistral** | XQuAD | 57.1 | 48.5 | — | 64.3 | 54.1 | — | 55.6 |
| | MGSM | 46.0 | 21.2 | 26.0 | 31.6 | 31.2 | 21.6 | 26.3 |
| | X-CSQA | 61.7 | 40.0 | 40.4 | 47.1 | 45.7 | 14.1 | 37.5 |
| | XLSum | 13.5 | — | 15.2 | 56.4 | 10.6 | 21.0 | 25.8 |

# F   Prompts

Table 10 shows the zero-shot prompts for each dataset. Note that when conducting tests in other languages, prompts are translated into the respective languages.

# G   Original Performance

Table 11 shows the original performance of Vanilla and Mistral on four tasks.

# H   Hyper-parameters

We adopt the performance on XQuAD in Chinese as the validation set to all languages and all tasks. Specifically, Table 12 shows the result on Vicuna when deactivating language-specific neurons in the understanding layer ($D_\mathcal{U}$) and generation layer ($D_\mathcal{G}$), where $N_1$ is the number of understanding layers and $N_2$ is the number of generation layer. We find that when setting $N_1 = 8$ and $N_2 = 2$, performance in English is influenced the least while performance in Chinese decreases the most. As for Mistral, the number is $N_1 = 6$ and $N_2 = 3$.

Table 12: XQuAD with Chinese on Vicuna.

| Method | $N_1$ | $D_\mathcal{U}$ ACC | $N_2$ | $D_\mathcal{G}$ ACC |
|--------|-------|----------|-------|----------|
| En-Vanilla | | 57.5 | | |
| Zh-Vanilla | | 55.5 | | |
| En-Deact | 8 | **57.7** (↑ 0.2) | 4 | 54.7 (↓ 2.8) |
| Zh-D-Deact | | **44.9** (↓ 10.6) | | 54.6 (↓ 0.9) |
| En-Deact | 6 | 58.6 (↑ 1.1) | 3 | 57.7 (↑ 0.2) |
| Zh-Deact | | 55.1 (↓ 0.4) | | 54.5 (↓ 1.0) |
| En-Deact | 4 | 57.3 (↓ 0.2) | **2** | **58.4** (↑ 0.9) |
| Zh-Deact | | 53.9 (↓ 1.6) | | **54.1** (↓ 1.4) |

Table 13: XQuAD with Chinese on Mistral.

| Method | $N_1$ | $D_\mathcal{U}$ ACC | $N_2$ | $D_\mathcal{G}$ ACC |
|--------|-------|----------|-------|----------|
| En-Vanilla | | 57.1 | | |
| Zh-Vanilla | | 64.3 | | |
| En-Deact | 8 | 53.3 (↓ 3.8) | 4 | 55.8 (↓ 1.3) |
| Zh-Deact | | 52.6 (↓ 11.7) | | 62.9 (↓ 1.4) |
| En-Deact | **6** | **56.8** (↓ 0.3) | **3** | **56.3** (↓ 0.8) |
| Zh-Deact | | **54.9** (↓ 9.4) | | **62.7** (↓ 1.6) |
| En-Deact | 4 | 57.6 (↑ 0.5) | 2 | 55.7 (↓ 1.4) |
| Zh-Deact | | 61.8 (↓ 2.5) | | 63.8 (↓ 0.5) |

# I Detailed Experiment Results

## I.1 Detailed Experiment Settings

**Reasoning Task** Deactivation methods: (i) randomly sampled neurons in the attention structure of task-solving layer. (ii) randomly sampled neurons in the task-solving layer. (iii) randomly sampled neurons in all layers. (iv) language-specific neurons in the task-solving layer. (v) language-specific neurons in the understanding layer and generation layer. (vi) language-specific neurons not in the attention structure of task-solving layers.

**Knowledge Question Answering Task** Deactivation methods: (i) randomly sampled neurons in the feed-forward structure of task-solving layers. (ii) randomly sampled neurons in the task-solving layer. (iii) randomly sampled neurons in all layers. (iv) language-specific neurons in the attention structure of task-solving layers. (v) language-specific neurons in the feed-forward structure of task-solving layers.

**Generation Task** Deactivation methods: (i) randomly sampled neurons in the generating layers. (ii) randomly sampled neurons in all layers. (iv) language-specific neurons in the generating layers.

## I.2 Detailed Result

Due to the limited space, we employ a more concise notation. We denote deactivating neurons in the self-attention layer of the $i$-th layer as $D_i^{(A)}$, while deactivating neurons in the feed-forward layer of the $i$-th layer is denoted as $D_i^{(F)}$. We denote $\mathcal{U} = \{1, \cdots, N_1\}$ as the set of layers that take charge of understanding as shown in Figure 2. Similarly, we denote $\mathcal{S} = \{N_1 + 1, \cdots, N_2\}$ as the set of layers that take charge of task solving and $\mathcal{G} = \{N_2 + 1, \cdots, 32\}$ as the set of layers that take charge of generation[5]. Furthermore, $D_{\mathcal{U}}^{(A)}$ represents deactivating neurons in self-attention layers of $\mathcal{U}$. Similarly, we introduce $D_{\mathcal{U}}^{(F)}$, $D_{\mathcal{S}}^{(A)}$, $D_{\mathcal{S}}^{(F)}$, $D_{\mathcal{G}}^{(A)}$ and $D_{\mathcal{G}}^{(A)}$.

Table 14: Understanding task.

| | Method | German | | | | Chinese | | | | Spanish | | | |
|---|---|---|---|---|---|---|---|---|---|---|---|---|---|
| | | En-D | De-D | $\Delta_{\text{En-D}}$ | $\Delta_{\text{De-D}}$ | En-D | Zh-D | $\Delta_{\text{En-D}}$ | $\Delta_{\text{Zh-D}}$ | En-D | Es-D | $\Delta_{\text{Es-D}}$ | $\Delta_{\text{Es-D}}$ |
| **Vicuna** | $D_{\mathcal{U}}^{R}$ | 57.8 | 49.7 | +0.3 | −0.6 | 57.8 | 55.8 | +0.3 | +0.1 | 57.8 | 56.1 | +0.3 | +0.4 |
| | $D_{All}^{R}$ | 57.9 | 50.8 | +0.4 | +0.5 | 57.9 | 55.8 | +0.4 | +0.1 | 57.9 | 55.9 | +0.4 | +0.2 |
| | $D_{\mathcal{U}}$ | 55.7 | 40.7 | −2.0 | −9.6 | 57.7 | 44.9 | +2.0 | −10.8 | 56.1 | 52.4 | −1.4 | −3.2 |
| | $D_{\mathcal{S}}$ | 48.3 | 41.7 | −7.2 | −8.6 | 45.0 | 45.4 | −12.5 | −10.3 | 29.5 | 28.6 | −28.0 | −27.1 |
| | $D_{\mathcal{G}}$ | 57.5 | 50.1 | 0.0 | −0.2 | 58.4 | 54.1 | +0.9 | −1.6 | 57.7 | 54.1 | +0.2 | −1.6 |
| **Mistral** | $D_{\mathcal{U}}^{R}$ | 58.1 | 48.2 | +1.0 | −0.4 | 58.1 | 63.9 | +1.0 | −0.4 | 58.1 | 54.3 | +1.0 | +0.2 |
| | $D_{All}^{R}$ | 57.6 | 48.3 | +0.5 | −0.3 | 57.6 | 63.6 | +0.5 | −0.7 | 57.6 | 54.5 | +0.5 | +0.4 |
| | $D_{\mathcal{U}}$ | 56.5 | 42.4 | −0.6 | −6.2 | 56.8 | 54.9 | −0.3 | −9.4 | 55.4 | 47.5 | −1.7 | −6.6 |
| | $D_{\mathcal{S}}$ | 54.3 | 43.2 | −2.8 | −5.4 | 54.9 | 52.9 | −2.2 | −11.4 | 50.3 | 44.9 | −6.8 | −9.2 |
| | $D_{\mathcal{G}}$ | 56.7 | 47.9 | −0.4 | −0.7 | 56.3 | 62.7 | −0.8 | −1.6 | 56.2 | 53.2 | −0.9 | −0.8 |

---

[5]Vicuna-7b-v1.5 and Mistral-7b-v1.0 both have 32 layers.

Table 15: Reasoning task.

| Method | German | | | | French | | | | Chinese | | | | Spanish | | | | Russian | | | |
|---|---|---|---|---|---|---|---|---|---|---|---|---|---|---|---|---|---|---|---|---|
| | En-D | De-D | $\Delta_{\text{En-D}}$ | $\Delta_{\text{De-D}}$ | En-D | Fr-D | $\Delta_{\text{En-D}}$ | $\Delta_{\text{Fr-D}}$ | En-D | Zh-D | $\Delta_{\text{En-D}}$ | $\Delta_{\text{Zh-D}}$ | En-D | Es-D | $\Delta_{\text{Es-D}}$ | $\Delta_{\text{Es-D}}$ | En-D | Ru-D | $\Delta_{\text{En-D}}$ | $\Delta_{\text{Ru-D}}$ |
| **Vicuna** $D_{S(A)}^R$ | 20.0 | 12.4 | −0.4 | −2.4 | 20.0 | 13.6 | −0.4 | −1.2 | 20.0 | 13.2 | −0.4 | +0.4 | 20.0 | 12.4 | −0.4 | −0.8 | 20.0 | 4.8 | −0.4 | −5.2 |
| $D_{S}^R$ | 18.4 | 12.4 | −2.0 | −2.4 | 18.4 | 14.0 | −2.0 | −0.8 | 18.4 | 14.4 | −2.0 | +1.6 | 18.4 | 15.2 | −2.0 | +2.0 | 18.4 | 4.8 | −2.0 | −5.2 |
| $D_{All}^R$ | 19.6 | 14.0 | −0.8 | −0.8 | 19.6 | 13.8 | −0.8 | −1.0 | 19.6 | 14.8 | −0.8 | +2.0 | 19.6 | 12.4 | −0.8 | −0.8 | 19.6 | 7.6 | −0.8 | −2.4 |
| $D_{S}$ | 3.6 | 2.0 | −16.8 | −12.8 | 8.4 | 3.2 | −12.0 | −11.6 | 4.8 | 4.0 | −15.6 | −8.8 | 8.8 | 4.0 | −11.6 | −9.2 | 10.4 | 4.0 | −10.0 | −6.0 |
| $D_{U\&G}$ | 16.4 | 5.6 | −4.0 | −9.2 | 19.2 | 9.6 | −1.2 | −5.2 | 20.0 | 9.2 | −0.4 | −3.6 | 17.6 | 11.6 | −2.8 | −1.6 | 17.2 | 5.6 | −3.2 | −4.4 |
| $\bar{D}_{S(A)}$ | 16.8 | 4.4 | −3.6 | −10.4 | 19.6 | 8.8 | −0.8 | −4.4 | 21.6 | 9.6 | +1.2 | −3.2 | 19.6 | 10.4 | −0.8 | −2.8 | 17.2 | 5.6 | −3.2 | −4.4 |
| **Mistral** $D_{S(A)}^R$ | 40.8 | 18.0 | −5.2 | −3.2 | 40.8 | 25.6 | −5.2 | −0.4 | 40.8 | 24.0 | −5.2 | −7.6 | 40.8 | 29.2 | −5.2 | −2.0 | 40.8 | 20.4 | −5.2 | −1.2 |
| $D_{S}^R$ | 39.2 | 20.0 | −6.8 | −1.2 | 39.2 | 25.2 | −6.8 | −0.8 | 39.2 | 25.6 | −6.8 | −6.0 | 39.2 | 29.6 | −6.8 | −1.6 | 39.2 | 19.6 | −6.8 | −2.0 |
| $D_{All}^R$ | 45.2 | 24.0 | −0.8 | +2.8 | 45.2 | 27.6 | −0.8 | +1.6 | 45.2 | 31.2 | −0.8 | −0.4 | 45.2 | 30.4 | −0.8 | −0.8 | 45.2 | 20.8 | −0.8 | −0.8 |
| $D_{S}$ | 38.4 | 12.0 | −7.6 | −9.2 | 40.8 | 24.8 | −5.2 | −1.2 | 37.9 | 19.6 | −8.1 | −12.0 | 40.4 | 24.4 | −5.6 | −6.8 | 33.6 | 11.2 | −12.4 | −10.4 |
| $D_{U\&G}$ | 42.4 | 9.2 | −3.6 | −12.0 | 41.2 | 21.6 | −4.8 | −4.4 | 46.4 | 19.6 | +0.4 | −12.0 | 44.0 | 28.0 | −2.0 | −3.2 | 46.0 | 12.0 | +0.0 | −9.6 |
| $\bar{D}_{S(A)}$ | 43.6 | 9.6 | −2.4 | −11.6 | 44.8 | 19.2 | −1.2 | −6.8 | 46.4 | 18.8 | +0.4 | −12.8 | 47.6 | 27.6 | +1.6 | −3.6 | 48.4 | 16.4 | +2.4 | −5.2 |

Table 16: Knowledge Question Answering task.

| Method | German | | | | French | | | | Chinese | | | | Spanish | | | | Russian | | | |
|---|---|---|---|---|---|---|---|---|---|---|---|---|---|---|---|---|---|---|---|---|
| | En-D | De-D | $\Delta_{\text{En-D}}$ | $\Delta_{\text{De-D}}$ | En-D | Fr-D | $\Delta_{\text{En-D}}$ | $\Delta_{\text{Fr-D}}$ | En-D | Zh-D | $\Delta_{\text{En-D}}$ | $\Delta_{\text{Zh-D}}$ | En-D | Es-D | $\Delta_{\text{Es-D}}$ | $\Delta_{\text{Es-D}}$ | En-D | Ru-D | $\Delta_{\text{En-D}}$ | $\Delta_{\text{Ru-D}}$ |
| **Vicuna** $D_{S(F)}^R$ | 57.5 | 43.8 | −0.3 | +0.0 | 57.5 | 40.3 | −0.3 | +0.2 | 57.5 | 43.2 | −0.3 | +0.0 | 57.5 | 44.6 | −0.3 | +0.3 | 57.5 | 25.5 | −0.3 | −0.5 |
| $D_{S}^R$ | 56.0 | 44.0 | −1.8 | +0.2 | 56.0 | 38.6 | −1.8 | −1.5 | 56.0 | 43.4 | −1.8 | +0.2 | 56.0 | 43.5 | −1.8 | −0.8 | 56.0 | 24.0 | −1.8 | −2.0 |
| $D_{All}^R$ | 57.7 | 43.6 | −0.1 | −0.2 | 57.7 | 40.5 | −0.1 | +0.4 | 57.7 | 43.2 | −0.1 | +0.0 | 57.7 | 44.5 | −0.1 | +0.2 | 57.7 | 26.0 | −0.1 | +0.0 |
| $D_{S(A)}$ | 34.8 | 43.4 | −23.0 | −0.4 | 32.6 | 31.1 | −25.2 | −12.7 | 32.6 | 28.9 | −25.2 | −14.3 | 20.4 | 25.0 | −37.1 | −19.3 | 48.3 | 22.9 | −9.5 | −3.1 |
| $D_{S(F)}$ | 57.8 | 41.5 | +0.0 | −2.5 | 57.2 | 37.8 | −0.6 | −6.0 | 56.9 | 39.6 | −0.9 | −3.6 | 57.6 | 43.0 | −0.2 | −1.3 | 57.8 | 25.6 | +0.0 | −0.4 |
| **Mistral** $D_{S(F)}^R$ | 61.0 | 40.2 | −0.7 | +0.2 | 61.0 | 40.1 | −0.7 | −0.3 | 61.0 | 46.7 | −0.7 | −0.4 | 61.0 | 45.2 | −0.7 | −0.5 | 61.0 | 12.7 | −0.7 | −1.4 |
| $D_{S}^R$ | 60.7 | 40.4 | −1.0 | +0.4 | 60.7 | 36.9 | −1.0 | −3.5 | 60.7 | 46.9 | −1.0 | −0.3 | 60.7 | 46.3 | −1.0 | +0.7 | 60.7 | 11.1 | −1.0 | −3.0 |
| $D_{All}^R$ | 61.8 | 40.1 | +0.1 | +0.1 | 61.8 | 40.7 | +0.1 | +0.3 | 61.8 | 47.2 | +0.1 | +0.1 | 61.8 | 44.7 | +0.1 | −1.0 | 61.8 | 14.1 | +0.1 | +0.0 |
| $D_{S(A)}$ | 50.4 | 32.3 | −11.3 | −7.7 | 55.3 | 27.4 | −6.4 | −13.0 | 54.7 | 42.4 | −7.0 | −4.7 | 44.5 | 34.1 | −17.2 | −11.6 | 51.1 | 8.3 | −10.6 | −5.8 |
| $D_{S(F)}$ | 61.5 | 38.1 | −0.2 | −1.9 | 61.2 | 38.1 | −0.5 | −2.3 | 61.3 | 43.5 | −0.4 | −3.6 | 61.0 | 43.9 | −0.7 | −1.8 | 60.8 | 11.8 | −0.4 | −2.3 |

Table 17: Generation task.

| Method | French | | | | Chinese | | | | Spanish | | | | Russian | | | |
|---|---|---|---|---|---|---|---|---|---|---|---|---|---|---|---|---|
| | En-D | Fr-D | $\Delta_{\text{En-D}}$ | $\Delta_{\text{Fr-D}}$ | En-D | Zh-D | $\Delta_{\text{En-D}}$ | $\Delta_{\text{Zh-D}}$ | En-D | Es-D | $\Delta_{\text{Es-D}}$ | $\Delta_{\text{Es-D}}$ | En-D | Ru-D | $\Delta_{\text{En-D}}$ | $\Delta_{\text{Ru-D}}$ |
| **Vicuna** $D_{G}^R$ | 13.2 | 14.2 | +0.1 | +0.0 | 13.2 | 61.6 | +0.1 | +0.5 | 13.2 | 10.4 | +0.1 | +0.0 | 13.2 | 20.8 | +0.1 | +0.0 |
| $D_{All}^R$ | 13.0 | 14.1 | −0.1 | −0.1 | 13.0 | 61.6 | −0.1 | +0.5 | 13.0 | 10.4 | −0.1 | +0.0 | 13.0 | 20.8 | −1.0 | +0.0 |
| $D_{G}$ | 13.0 | 13.8 | −0.1 | −0.4 | 13.1 | 59.5 | +0.0 | −1.6 | 13.0 | 9.1 | −0.1 | −1.3 | 13.1 | 20.3 | +0.0 | −0.5 |
| **Mistral** $D_{G}^R$ | 13.6 | 15.2 | +0.1 | +0.0 | 13.6 | 56.7 | +0.1 | +0.3 | 13.6 | 10.3 | +0.1 | −0.3 | 13.6 | 21.2 | +0.1 | +0.2 |
| $D_{All}^R$ | 13.6 | 15.4 | +0.1 | +0.2 | 13.6 | 55.9 | +0.1 | −0.5 | 13.6 | 10.2 | +0.1 | −0.4 | 13.6 | 21.1 | +0.1 | +0.1 |
| $D_{G}$ | 14.3 | 14.2 | +0.8 | −1.0 | 13.6 | 52.8 | +0.1 | −3.6 | 13.7 | 10.2 | +0.2 | −0.4 | 13.5 | 20.2 | −0.1 | −0.8 |

