# OpenReview forum: "How do Large Language Models Handle Multilingualism?"
_NeurIPS.cc/2024/Conference — NeurIPS 2024 poster_

### Official Review · Reviewer_NLiQ · 2024-07-13

**Soundness:** 3
**Presentation:** 4
**Contribution:** 4
**Rating:** 7
**Confidence:** 3

**Summary:**

This paper delves deeper into how LLMs handle multilingualism. The authors hypothesized a three-stage multilingual workflow called MWork: understanding, task-solving, and generating and which language(s) become essential in each stage. To verify their proposed workflow, they experimentally identify language-specific parameters and selectively deactivate them within different structures so this allows to assess the functionality of corresponding structures and enables the hypothesis. To do so, the authors develop a novel approach called Parallel Language-specific Neuron Detection (PLND). Without requiring labeled data, PLND can identify the language specific neurons. Following the PLND, the authors successfully identify the language-specific neurons which account for only 0.13% of all neurons. Their extensive results report that, by deactivating those neurons, the multilingual task performance could significantly dropped.

This paper tackles an important research question of LLMs: how do large language models handle multilingualism? To address this, they deliberately design a new approach of PLND and successfully identifies a language-specific neurons in the LLM models. Along with their MWork hypothesis, they carefully conducted experiments with multiple languages with different scrips and verified their hypothesized three-stage multilingual workflow.

This paper develops a useful model analysis tool of PLND. Their hypothesis is well explored in multiple languages. Technically well sound.

**Strengths:**

- well-organized paper. clear presentation.
- Technically sound by extensive experiments with different natural language understanding tasks with diverse language sets

**Weaknesses:**

- no major weakness though

**Questions:**

- Regarding those language-specific neuron, are there any patterns or trends related to language family grouping or similarity? Such analysis or findings would be interesting to NLP researchers.
-  Have you ever tried this approach against bigger models with > 7B parameters?
- Since language-specific neurons are identified via the proposed framework, can we do model compression by preserving those specific parameters without losing multilingual capability? If you have examined this perspective, can you please share the results?

---

> ### Author Rebuttal · Authors · 2024-08-07
>
> Dear Reviewer NLiQ,
>
>
>
> Thank you for your insightful reviews and comments. We appreciate the time and effort you have put into providing valuable feedback. We would like to address your concerns as follows:
>
>
>
> > Question #1: Patterns of language-specific neurons.
>
> We acknowledge your concerns regarding the properties of language-specific neurons, which have been thoroughly investigated in Appendix C (lines 512 to 523). Given its significance, we are considering integrate this section into the main text in the final version. Thanks for your recommendation.
>
>
>
> > Question #2: Performance on larger models.
>
> Thank you for recommending adding analysis on larger models. Regrettably, we are unable to implement our methods on larger models at this time due to resource constraints. In line with Reviewer TbKY's suggestion, we will adjust the claim "all LLMs" accordingly in the final version.
>
>
>
> > Question #3: Model compression.
>
> We appreciate your invaluable insights on the model compression potential of our proposed method. We also explore this direction following this paper.
>
> We first take the opposite approach to language-dependent neurons. Specifically, we extract neurons that are not important for all corpora, denoted as Language-Irrelevant Neutons. We conduct testing in Thai on Mistral-7b-base, employing XLSum as our chosen task. Opting for XLSum is driven by its simplicity in contrast to complex NLP tasks like MGSM, yet it effectively showcases the model's multilingual ability when compared to measuring perplexity. Our analysis reveals that these language-irrelevant neutons represent a mere fraction of the overall parameters (approximately 2%), with model performance showing negligible impact upon deactivation. In contrast, random deactivation of 0.16B parameters (equivalent to 2% of all parameters) markedly impairs the model's functionality.
>
> | # Deactivated Parameter | Language-Irrelevant Neutons | Random |
> | ----------------------- | --------------------------- | ------ |
> | 0.11B                   | 24.3                        | 23.7   |
> | 0.16B                   | 24.6                        | <1     |
>
>
>
> Nevertheless, the modest compression ratio proves unnecessary, and we think it's because the filtering criteria for language-irrelevant neurons are too strict. Consequently, we opt to retain language-specific neurons. Specifically, we implement language-specific neurons and deactivate non-language-specific neurons, resulting in improved performance compared to our previous setup. We can achieve nearly unaffected performance by deactivating up to 23% of all parameters. The detailed performance breakdown is presented below.
>
> | # Deactivated Parameter | 0.26B | 0.52B | 0.79B | 1.1B | 1.3B | 1.6B |
> | ----------------------- | ----- | ----- | ----- | ---- | ---- | ---- |
> | Performance             | 23.7  | 24    | 24.9  | 24.5 | 23.3 | 23.1 |

---

> ### Author Response · Authors · 2024-08-11
> **Initial findings for your suggested future directions**
>
> Dear Reviewer NLiQ,
>
> I'm writing to express our gratitude for the time and effort you've dedicated to reviewing our paper. Your insightful questions have truly illuminated new directions for exploration. Please kindly refer to our first response for some preliminary results.
>
> As the discussion period nears its end, we would be delighted to receive any further comments or discussions you might have. Your input would be invaluable in enhancing our work and inspiring further research ideas.
>
> Thank you once again for your thoughtful review and support.
>
> Warm regards,
> Authors

---

> > ### Comment · Reviewer_NLiQ · 2024-08-13
> >
> > Thank you for the clarification and the additional experimental results. This paper presents extensive experiments with interesting findings. I have read all the reviews and the corresponding authors' responses, and I don't see any major concerns. Therefore, I will keep my score as it is  (7: accept).

---

### Official Review · Reviewer_TbKY · 2024-07-13

**Soundness:** 3
**Presentation:** 4
**Contribution:** 4
**Rating:** 6
**Confidence:** 4

**Summary:**

The paper presents two key contributions. One is Parallel Language-specific Neuron Detection, a method of identifying elements of multilingual LLMs that are responsible for handling particular languages; the method only requires unlabeled text data in each language in order to detect these neurons, which makes it cheap and efficient. The second contribution is an insight into the workflow of multilingual LLMs (referred to in the paper as MWork), claiming that LLMs process input in three steps: understanding, task solving and output generation; task solving further splits into retrieval and thinking. Also, the thinking step is shown to be happening in English, while understanding, knowledge retrieval and generation are handled multilingually / specifically to the language of the input. Authors use PLND to verify that the models chosen for the experiments actually follow this workflow and conduct thorough experiments on two models about a dozen languages other than English, including high-resource and low-resource examples.

**Strengths:**

- Both key contributions (PLND and MWork) are of fundamental relevance to the field of LLM interpretability and analysis

- Presented experiments are rigorous and thorough

- The paper is written clearly and although the subject is complex and multi-faceted, the authors do a good job of introducing the necessary concepts and definitions and explaining the rationale behind their work as well as the conducted experiments and analysis of their results.

**Weaknesses:**

W1. Most importantly, the claims are very broad ("LLMs are this / LLMs do that"), however this is verified on two very similar models: the included Mistral and Vicuna models both have 7 billion parameters, which puts them on the modest end of modern LLMs. How can we be sure that models with more parameters (13 / 30 / 70 / ...) behave in a similar manner without testing?, what about the influence of the context length on the performance? This is even mentioned in the limitations section of the paper: I am not suggesting that the authors run an additional set of expensive experiments -- I am suggesting that the paper text should be adjusted to reflect the actual findings of the presented experiments and would avoid bold claims about "all LLMs".

W2. Another weakly supported claim is that 400 documents was enough for language-specific neuron tuning, simply because tuning on 800 documents did not yield better results. I find that this is a single instance and without running more comparisons and more thorough comparisons this claim does not hold and should be adjusted in the text. This is especially important since improvements of 2.3% (low-resource) and 3.6% (high-resource) are small-scaled improvements and bigger amounts of language-specific
 data could be expected to yield bigger improvements. Again, rather than running tons of more experiments for a single paper I instead suggest that the paper text should be adjusted (e.g. calling it a "pilot study" or "preliminary indications/results" -- PLND and MWork themselves are already worthy contributions).

W3. Not a strong weakness, but minor typos:
- row 101: "With a set of n corpus" --> "With a set of n corpora", same on row 103 ('corpus' in plural is 'corpora')
- row 319: "one brunch of work" --> "one branch of work"

**Questions:**

Q1. There seems to be a leap from the experimental results to conclusions about how LLMs behave. Are the presented conclusions (about understanding->thinking->generating) the only possible explanation of the obtained results, or could there be other explanations?

Q2. In your work the language-specificity is defined via that language only (importance is higher than a threshold). Would you expect a stronger performance of PLND if the language-specificity were defined contrastively, that is high impact on one language and low impact on other languages?, in other words, could a neuron be important to several languages?, and what would be the repercussions of such neurons on your work.

Q3. In this work you focus on the "English / non-English" setting. Is this driven by the fact that English is the most abundant language in the training data?, and what would you expect the behavior of the models to be if another language were the most frequent, would the thinking be done in that language?, what about a more balanced setting where more than one language are dominant in the data (for example 40% English, 40% Chinese and other less frequent languages)?

Q4. Concerning language-specific tuning, hypothetically, is multilingual performance of LLMs bound by English-centerdness? that is, can one achieve better performance of an LLM in a non-English language if that language's data is included into pre-training from the start, rather than being English-centered and tuned to a non-English language?

Q5. Modern language models are often trained with programming code added to natural language texts. What would you expect your analysis to show in case the input is programming code or comments?

Q6. Could full fine-tuning to a non-English language achieve better results than just tuning that language's language-specific neurons?

**Limitations:**

All good.

---

> ### Author Rebuttal · Authors · 2024-08-07
>
> Dear Reviewer TbKY,
>
>
>
> Thank you for your insightful reviews and comments. We appreciate the time and effort you have put into providing valuable feedback. We would like to address your concerns as follows:
>
>
>
> > Concern #1: Overclaim on model size and multilingual enhancement experiments.
>
> We appreciate your concern regarding the claims including "LLMs xxxx" and "enhancing by 400 documents". We acknowledge that the claim for "all LLMs" is strong and will revise that accordingly in the final version. Regarding the the enhancement experiment, we conduct experiments on tuning language-specific neurons by more documents from Wikipedia and here are detailed results.
>
> |      | **Original** | **0.4k** | **1k** | **8k** | **64k** | **128k** |
> | ---- | ------------ | -------- | ------ | ------ | ------- | -------- |
> | Vi   | 32.7         | **34.9** | 34.3   | 28.3   | 26.4    | 22.1     |
> | Th   | 25.6         | **28.5** | 27.4   | 23.7   | 20.8    | 15.9     |
> | Ar   | 21.7         | **23.4** | 22.8   | 19.2   | 16.2    | 8.3      |
> | Sw   | 15.1         | **16.9** | 15.2   | 12.4   | 7.9     | 5.5      |
>
> We find that adding more documents significantly reduces performance. This phenomenon is likely attributed to overfitting the model to Wikipedia data, causing conflicts with patterns in the newly introduced data. To further validate our assumption and explore the effectiveness of language-specific tuning, we conduct additional experiments by assembling the training set from a wider array of resources such as textbook, website and journalistic materials. Due to space limitations, please kindly refer to the rebuttal addressing concern #2 provided to Reviewer 2b8j for further details.
>
>
>
> > Question #1: Other assumption of framework based on observation in Figure 1
>
> We would like to share more details of our preliminary exploration, which involves analyzing the content of embeddings but is removed from the paper for the sake of coherence. Figure (1) in the attached PDF illustrates the decoded embedding of various layers of Vicuna-13b processing the Chinese version of "What is one plus one equal to?". Through this examination, we observe a progression where the model initially comprehends the input, transitioning tokens to English such as "equal" and "?", in the early layers. Subsequently, in the intermediate layers, nearly all tokens are in English, indicating the task solving in English. Finally, in the later layers, the English answer is transferred back into Chinese. In summary, our proposed framework draws inspiration not just from token distribution but also from a thorough examination of token meanings.
>
> > Question #2: Could a neuron be important to several languages?
>
> The answer is yes, and we have investigated the degree of overlap among language-specific neurons in Appendix C (line 512 to line 523).
>
>
>
> > Question #3 & #4: What is the performance of non-English-centered LLMs.
>
> We appreciate your concerns on other types of LLMs. Our analysis employs Qwen2-7b, a model that is not centered around English due to the comparable size of training data in Chinese and English. In Figure (2) in attached PDF, the distribution of tokens across layers is depicted. We observe that Qwen2 tends to quickly transition multilingual inputs to Chinese or English to address tasks predominantly in English, albeit with some reliance on Chinese support. Towards the end, answers in English or Chinese are reverted back to multilingual languages.
>
> > Question #5: Whether can implement on code?
>
> We appreciate your invaluable insights on the generalibity of our proposed method. We also explore this direction following this paper. Our initial experiments indicate its potential applicability in code enhancement. Specifically, we utilize an instruction-code dataset [3] to identify code-specific neurons and subsequently fine-tune these specialized neurons. Experiments are conducted on Llama3-8b-Instruct and here are detailed results. Remarkably, we observe that training on a corpus of only 6k instances for less than 10 minutes can enhance the model's coding capabilities without compromising, and in some cases even enhancing, other aspects of the model. This outcome may be attributed to a clearer division of responsibilities among the parameters, culminating in superior overall model performance.
>
> |          | Human-Eval | MGSM     |
> | -------- | ---------- | -------- |
> | Original | 32.6       | 56.4     |
> | 2k       | 35.6       | 57.2     |
> | 6k       | **37.8**   | **61.6** |
>
>
>
> > Question #6: Could full fine-tuning performs better than language-specific tuning?
>
> Full fine-tuning demands a substantial amount of training data and may diminish performance in languages like English and Chinese, as evidenced by other continue-training works focusing on multilinguals [1]. Some works merge English corpus in multilingual training data to keep its English performance either in continue train or sft [2] [4]. However, these drawbacks can be avoided by our language-specific neuron tuning method.
>
>
>
>
>
> [1] Sailor: Open Language Models for South-East Asia, Arxiv 2024
>
> [2] SeaLLMs - Large Language Models for Southeast Asia, ACL 2024
>
> [3] Python_code_instructions_18k_alpaca
>
> [4] Multilingual Instruction Tuning With Just a Pinch of Multilinguality, ACL 2024

---

> ### Author Response · Authors · 2024-08-11
> **Follow-up on Our Rebuttal Submission**
>
> Dear Reviewer TbKY,
>
> I hope this message finds you well. We are grateful for your valuable feedback on our submission and are pleased to see your positive score. We have addressed the points you raised in detail in our responses.
>
> As the discussion period is coming to a close soon, we kindly ask if you could review our responses at your earliest convenience. We are eager to know if our explanations have alleviated your concerns. If there are still areas needing improvement, your insights would be greatly appreciated and instrumental in enhancing our work.
>
> Thank you once again for your thoughtful review and support.
>
> Warm regards,
> Authors

---

### Official Review · Reviewer_Nru9 · 2024-07-14

**Soundness:** 3
**Presentation:** 3
**Contribution:** 2
**Rating:** 5
**Confidence:** 4

**Summary:**

This paper proposes MWork, a workflow to study how large language models handle multilingual inputs. The main idea is to detect English and non-English neurons by probing their value differences and performance gaps. Based on the results, they argue that there are three steps in the workflow: understanding, task solving, and generating. They also use experiments to support their hypothesis.

**Strengths:**

- They propose an interesting way to understand the mechanism of large language models.

**Weaknesses:**

- The authors classify neurons based on English and non-English, which is not very convincing to me. I believe that, beyond these two categories, there should be some neurons that are helpful for **all** languages. These neurons can be used for general understanding and logical reasoning. However, this paper ignores this aspect. See more explanations below.
  -  For neuron detection, when computing the impact, I am curious why you do not remove the overlapping neurons. In this way, you can detect the real language-specific ones. Specifically, in Appendix C, although the authors argue that the intersection with English from other languages is relatively limited based on rows, if we look at the columns, it seems that all the language-specific neurons are covered by English ones, suggesting that they are actually not language-specific. It is very likely that these neurons are essential for general understanding and are language-agnostic.
  - It is not clear how English and non-English tokens are detected in Figure 1. Do you determine this by applying a decoder to decode from the hidden representations?
  - The authors argue that large language models will use English tokens for task solving with multilingual inputs based on the interpretation of Figure 1. However, Figure 1 also shows that models will use non-English tokens for task solving with English. I don’t think this is reasonable. From my perspective, the way to classify tokens or neurons is not accurate, and the ignorance of language-agnostic neurons makes this figure unreasonable.
  - From the experimental results, deactivating language-specific neurons sometimes causes a performance drop for both English and non-English, suggesting the existence of general neurons.

**Questions:**

See above.

**Limitations:**

Yes.

---

> ### Author Rebuttal · Authors · 2024-08-07
>
> Dear Reviewer Nru9,
>
>
> We appreciate the time and effort you have put into providing valuable feedback. However, we respectfully believe there is a serious misunderstanding regarding our work. We would appreciate the opportunity to clarify a few points and address your concerns as follows.
>
>
> > Misunderstanding: Exist language-agnostic neurons.
>
>
> The proposed "language-specific neuron" and the framework do not contradict the presence of language-agnostic neurons. In reality, language-specific neurons depend on these language-agnostic neurons to complete latent comprehensive understanding and logical reasoning. Language-agnostic neurons reside in neurons not identified as language-specific. Analogously to humans, when an English native speaker encounters content in other languages, the solution lies in their overall understanding and problem-solving skills, surpassing any linguistic barrier.
>
> It is crucial to note that language-specific neurons constitute merely 0.1% of all neurons for each non-English language (nearly 0.3% for English). Without the presence of language-agnostic neurons, compressing the model by a factor of 1000 and retaining only language-specific neurons will maintain performance. However, it is obviously impossible. These definitions and observations related to language-agnostic neurons also align with those from concurrent works [1, 2].
>
> Although they account for the majority of neurons, language-agnostic neurons are not the focus of this paper. We mainly investigate how LLMs leverage the English language's capabilities for handling multilingualism. Our findings indicate that LLMs predominantly utilize neurons as language-agnostic entities for comprehension and reasoning, reserving only a select few to explicitly express their understanding and logic in diverse linguistic contexts comprehension and reasoning across various languages. This is exactly the motivation of our language-specific neuron enhancement method.
>
> Nevertheless, we acknowledge your concern and will include the discussion regarding language-agnostic neurons in the final version.
>
> > Question #1: Why do not remove overlapping of language-specific neurons.
>
> As explained earlier, what we discovered are actually language-specific neurons, constituting only a very small fraction of all neurons. This specific region is solely dedicated to language processing, distinct from general comprehension or reasoning abilities. This discovery is also consistent with concurrent research [3]. Therefore, it does not make sense to remove overlapped neurons as they are already specialized for language.
>
> Furthermore, neurons that overlap across all languages account for only 0.02% of all parameters. It is implausible that LLMs depend on this minute fraction of neurons for intricate comprehension and reasoning. Instead, these shared neurons between two or more languages merely denote multifunctionality, such as in an individual proficient in both French and Spanish concurrently, rather than being language-agnostic.
>
>
>
> > Question #2: How tokens are decoded in Figure 1?
>
> We employ decoder of the last layer to decode hidden representations, a method also employed in a concurrent work [4] that shares similar findings with ours, also interpreting the phenomenon as "the model thinks in English."
>
>
>
> > Question #3: Why Figure 1 contains non-English tokens?
>
> It should certainly contain non-English tokens; at the very least, non-English tokens are necessary to fill in entities although the model thinks in English. Additionally, we claim that the feed-forward structure of the task-solving layer is used to extract multilingual knowledge to obtain factual content. These extracted multilingual tokens necessitate neurons for processing and consequently manifest within the hidden representation.
>
>
>
> > Question #4: Why deactivating language-specific neurons influence English performance?
>
> As analyzed in the paper from Section 3.3 to Section 3.6, performance drop in English results from disabling the task-solving layer. If neurons are deactivated appropriately, English performance remains unaffected, as highlighted in Tables 2 to 6.
>
>
>
>
>
>
>
> [1] Language-Specific Neurons: The Key to Multilingual Capabilities in Large Language Models, ACL 2024
>
> [2] Multilingual Knowledge Editing with Language-Agnostic Factual Neurons, Arxiv 2024
>
> [3] Unveiling Linguistic Regions in Large Language Models, ACL 2024
>
> [4] Do Llamas Work in English? On the Latent Language of Multilingual Transformers, ACL 2024

---

> ### Author Response · Authors · 2024-08-11
> **Rebuttal Review Required for Accurate Assessment**
>
> Dear Reviewer Nru9,
>
> I hope this message finds you well. The discussion period is ending soon, I am writing to emphasize the importance of your review for our submission. Your score is significantly lower than the other three reviewers, and we believe this discrepancy may indicate a misunderstanding or oversight.
>
> We have addressed all the concerns in our detailed rebuttal and would appreciate your prompt attention to it. A thorough reassessment is crucial to ensure a fair evaluation.
>
> Your expertise is highly valued, and we trust that a reconsidered review will reflect the true merit of our work.
>
> Thank you for your immediate attention to this matter.
>
> Best regards,
> Authors

---

> > ### Comment · Reviewer_Nru9 · 2024-08-11
> >
> > Thank you for your response. I would like to clarify my concerns.
> >
> > **Regarding language-agnostic neurons:** When I mention "language-agnostic neurons," I'm referring to the "overlapping neurons" found across language-specific neurons. As you show in Appendix C, there is a certain degree of overlap among neurons across different languages, indicating that they are not entirely language-specific. I believe it would be more appropriate to exclude these neurons from the language-specific category, as they may play a crucial role in reasoning and task-solving. If these neurons are not removed, the performance drop observed when removing English-specific or non-English-specific neurons could primarily be due to the loss of these important task-solving neurons rather than actual language differences. This is why I am not fully convinced by the experimental results.

---

> > > ### Author Response · Authors · 2024-08-11
> > >
> > > Dear Reviewer Nru9,
> > >
> > > Thanks for further clarifying your concern. We acknowledge and value your observation regarding the potential language-agnostic nature of overlapped language-specific neurons and their significance in managing multilingualism.
> > >
> > > As detailed in our rebuttal, it is important to note that language-agnostic neurons primarily constitute those that are not identified as language-specific, encompassing approximately 99% of all neurons. As evidenced in Figure 5 and Figure 6 in Appendix C, while certain languages exhibit overlaps, most language pairs share only a small fraction of neurons (less than 0.5). Furthermore, upon calculating the neurons overlapped across all languages, they represent a mere 0.02% of the total neuronal population.  These shared neurons between two or more languages merely denote multifunctionality, such as in an individual proficient in both French and Spanish concurrently, rather than being language-agnostic.
> > >
> > > We are grateful for your feedback and are currently conducting a related experiment to provide empirical evidence. We will provide you with our findings once we get the results during the ongoing discussion phase.
> > >
> > > Best Regards,
> > > Author

---

> > > > ### Comment · Reviewer_Nru9 · 2024-08-12
> > > >
> > > > Thanks for the reply. I personally think 0.4~0.5 overlapping ratio is not a small number and can potentially affect the results a lot. Also, I believe even a small amount of neurons can contribute a lot to the model behavior, as the importance of neurons in language models are not equally distributed. It would be great if you can provide the experiments with excluding those overlapping neurons. I am looking forward to the new empirical evidence you will have. Thanks.

---

> > > > > ### Author Response · Authors · 2024-08-13
> > > > >
> > > > > Dear Reviewer Nru9,
> > > > >
> > > > >
> > > > >
> > > > > We appreciate your concern about the language-agnostic property of overlapped neurons among language-specific neurons across different languages. To address this, we have conducted the following experiments.
> > > > >
> > > > > We initially implement a radical deactivation approach, wherein we specifically deactivate overlapping elements between each language and English. These elements precisely correspond to the intersecting neurons in the first column of Figure 5. Presented below are the comprehensive findings pertaining to Mistral. Our evaluation is centered around the reasoning task, recognized as the most indicative and challenging assessment for the model.  We compare under the optimal "deactivating" method, which involves deactivating all language-specific neurons except those in S-ATTN.
> > > > >
> > > > > |                                        | Eng  | n-Eng | $\Delta_{Eng}$ | $\Delta_{n-Eng}$ | $\Delta\uparrow$ |
> > > > > | -------------------------------------- | ---- | ----- | -------------- | ---------------- | ---------------- |
> > > > > | All LSN                                | 46.2 | 18.3  | +0.2           | -8.0             | +8.2             |
> > > > > | LSN without overlapped between English | 45.8 | 20.2  | -0.2           | -6.1             | +5.9             |
> > > > >
> > > > > As evident, the performance of English remains stable, contrasting sharply with the significant decline in the performance of multilingual. Removing overlapped neurons, as opposed to deactivating all language-specific neurons, leads to a less pronounced drop, yet the impact remains noteworthy. This demonstrates that overlapped neurons are not language-agnostic; they are not utilized for general comprehension and logical reasoning. Otherwise, the fundamental reasoning capacity and performance in multilingual contexts would remain unaffected.
> > > > >
> > > > >
> > > > >
> > > > > In addition, we retained the language-specific neurons that overlapped in all languages, meaning that we removed them from the language-specific neurons to be deactivated. Detailed results follow.
> > > > >
> > > > > |                                      | Eng  | n-Eng | $\Delta_{Eng}$ | $\Delta_{n-Eng}$ | $\Delta\uparrow$ |
> > > > > | ------------------------------------ | ---- | ----- | -------------- | ---------------- | ---------------- |
> > > > > | All LSN                              | 46.2 | 18.3  | +0.2           | -8.0             | +8.2             |
> > > > > | LSN without all languages overlapped | 45.6 | 18.7  | -0.4           | -7.6             | +7.2             |
> > > > >
> > > > > As we explained in our previous rebuttal, the neurons that overlap across all languages only account for 0.02% of the total number of neurons. From the results, we can see that the performance is almost the same as deactivating all language-specific neurons. This further proves that these neurons are not language-agnostic neurons, but only a subset of language-specific neurons.
> > > > >
> > > > > Furthermore, if we extend the overlapping neurons to languages including real low-source languages such as Vietnamese, Thai, Arabic, and Swahili (as shown in Table 6), the overlapping neurons account for less than 1 in 100,000. We agree that the importance of neurons in a language model is not evenly distributed, as neurons in a specific language account for only 0.1% of all neurons. However, it is hard to believe that 1 in 100,000 neurons can bear such important responsibilities such as basic reasoning and understanding, and this proportion will further decrease as the number of languages increases.

---

> > > > > > ### Comment · Reviewer_Nru9 · 2024-08-13
> > > > > >
> > > > > > Thanks for the additional results. This provides more insight regarding how those neurons contribute to the performance. Please include them in the final version. I've modify my score accordingly.

---

### Official Review · Reviewer_2b8J · 2024-07-17

**Soundness:** 3
**Presentation:** 3
**Contribution:** 3
**Rating:** 7
**Confidence:** 5

**Summary:**

This paper examines how LLMs handle multilingualism. The author proposes a hypothetic workflow (MWork), which suggests that LLMs understand the multilingual query, think in English, and than generate results in the input language. A neuron detection method is proposed to detect language specific neurons. By deactivation some of the neurons, the authors validates the above workflow, and show that the  ability for a specific language could be improved by finetuning only language specific neurons.

**Strengths:**

The paper presents an interesting  hypothesized workflow and validate it with neuron analysis.

The analysis with neuron deactivation is sound enough.

**Weaknesses:**

The analysis of this paper is based on the understanding of  layers. However, it might be possible that different tasks or even different instances may have different splitting of layers for different stages in the workflow. How would this affect the analysis?

It is still strange to know that 400 documents could improve the language ability of a low-resource languages, especially when 800 documents will not be more helpful.

**Questions:**

See the weakness part.

---

> ### Author Rebuttal · Authors · 2024-08-07
>
> Dear Reviwer 2b8J,
>
>
>
> Thank you for your insightful reviews and comments. We appreciate the time and effort you have put into providing valuable feedback. We would like to address your concerns as follows:
>
>
>
> > Concern #1: different instances may have different splitting of layers
>
> We appreciate your concern regarding varying splitting settings for different tasks and different instances. Yes, the boundaries between layers are not very clear and vary across models, languages and tasks. In our paper, for each model we consistently apply the same splitting approach across various languages and tasks to demonstrate the generalizability of our proposed framework. It is reasonable to speculate that more precise and finely-tuned layer splitting could improve the performance of deactivation experiments (Table 2 to Table 5) and the enhancement experiments in Section 4.
>
> > Concern #2: why 400 documents can improve language ability
>
> We acknowledge your concern regarding why only 400 documents can yield satisfactory performance. When considering that language-specific neurons account for just 0.1% of all parameters, this size of training corpus seems reasonable. Other studies that boost specific low-resource languages [1] [2] necessitate nearly millions of documents, comparable with scaling the number of language-specific neurons proportionally.
>
>
>
> We conduct experiments on tuning language-specific neurons by more documents from Wikipedia and here are detailed results.
>
> |      | **Original** | **0.4k** | **1k** | **8k** | **64k** | **128k** |
> | ---- | ------------ | -------- | ------ | ------ | ------- | -------- |
> | Vi   | 32.7         | **34.9** | 34.3   | 28.3   | 26.4    | 22.1     |
> | Th   | 25.6         | **28.5** | 27.4   | 23.7   | 20.8    | 15.9     |
> | Ar   | 21.7         | **23.4** | 22.8   | 19.2   | 16.2    | 8.3      |
> | Sw   | 15.1         | **16.9** | 15.2   | 12.4   | 7.9     | 5.5      |
>
> We find that adding more documents significantly reduces performance. This phenomenon is likely attributed to overfitting the model to Wikipedia data, causing conflicts with patterns in the newly introduced data.
>
> To further validate our assumption and explore the effectiveness of language-specific tuning, we conduct additional experiments by assembling the training set from a wider array of resources such as textbook, website and journalistic materials. Employing the same settings as in Table 6, we present detailed results below. While 0.4k documents enhance multilingual proficiency, 1k and 8k documents disrupt this ability as new patterns conflict with those learned during pre-training. However, expanding the training corpus to 64k and 128k documents reconstructs multilingual proficiency and boosts overall performance.
>
> Our findings reveal that enhancing model's multilingual capability through neuron-specific continue-training undergoes a process of disruption and reconstruction. However, adding more data from Wikipedia fails to reconstruct the multilingual ability due to overfitting. Notably, the training data comprising 128k documents remains significantly lower—by a factor of 10 to 100—than the quantities required in previous studies [1] and [2].
>
> |      | **Original** | **0.4k** | **1k** | **8k** | **64k** | **128k** |
> | ---- | ------------ | -------- | ------ | ------ | ------- | -------- |
> | Vi   | 32.7         | 34.3     | 31.8   | 31.5   | 33.7    | **35.2** |
> | Th   | 25.6         | 27.8     | 25.4   | 25.8   | 26.5    | **30.7** |
> | Ar   | 21.7         | 22.6     | 21.3   | 20.1   | 22.4    | **24.9** |
> | Sw   | 15.1         | 16.4     | 15.2   | 16.0   | 16.6    | **17.3** |
>
>
>
> [1] Sailor: Open Language Models for South-East Asia, Arxiv 2024
>
> [2] SeaLLMs - Large Language Models for Southeast Asia, ACL 2024

---

> ### Author Response · Authors · 2024-08-11
> **Regarding why 400 documents yield good performance**
>
> Dear Reviewer 2b8J,
>
> I hope this message finds you well. We are grateful for the time and effort you have put into reviewing our paper. Your concern regarding why only 400 documents can yield satisfactory performance indeed motivated us to do an additional experiment, kindly refer to the later part of our first response.
>
> As the discussion period is coming to a close soon, we are eager to know if our explanations have alleviated your concerns. If there are still areas needing improvement, your insights would be greatly appreciated and instrumental in enhancing our work.
>
> Thank you once again for your thoughtful review and support.
>
> Warm regards, Authors

---

### Author Rebuttal · Authors · 2024-08-07

Thank you for your insightful reviews and comments. We include Figures added in rebuttal in the attached PDF.

---

### Decision · Program_Chairs · 2024-09-25

**Decision:**

Accept (poster)

**Comment:**

This paper proposes a neuron-level workflow for detecting the multilingual capabilities of LLMs. The motivation is interesting, and the experimental analysis is solid. During the rebuttal phase, the authors engaged in a profound discussion with the reviewers and effectively addressed most concerns. Given the intriguing insights this research offers into multilingual capabilities, I support its acceptance.